# A Non-Enzyme and Non-Label Sensitive Fluorescent Aptasensor Based on Simulation-Assisted and Target-Triggered Hairpin Probe Self-Assembly for Ochratoxin a Detection

**DOI:** 10.3390/toxins12060376

**Published:** 2020-06-06

**Authors:** Mengyao Qian, Wenxiao Hu, Luhui Wang, Yue Wang, Yafei Dong

**Affiliations:** 1School of Computer Science, Shaanxi Normal University, Xi’an 710119, China; qmy@snnu.edu.cn (M.Q.); wy.wangyue@snnu.edu.cn (Y.W.); 2College of Life Sciences, Shaanxi Normal University, Xi’an 710119, China; huwenxiao@snnu.edu.cn (W.H.); wangluhui@snnu.edu.cn (L.W.); 3National Engineering Laboratory for Resource Developing of Endangered Chinese Crude Drugs in Northwest of China, Shaanxi Normal University, Xi’an 710119, China

**Keywords:** ochratoxin A, fluorescence, G-quadruplex, biosensor, computation, simulation

## Abstract

The monitoring and control of mycotoxins has caused widespread concern due to their adverse effects on human health. In this research, a simple, sensitive and non-label fluorescent aptasensor has been reported for mycotoxin ochratoxin A (OTA) detection based on high selectivity of aptamers and amplification of non-enzyme hybridization chain reaction (HCR). After the introduction of OTA, the aptamer portion of hairpin probe H1 will combine with OTA to form OTA-aptamer complexes. Subsequently, the remainder of the opened H1 will act as an initiator for the HCR between the two hairpin probes, causing H1 and H2 to be sequentially opened and assembled into continuous DNA duplexes embedded with numerous G-quadruplexes, leading to a significant enhancement in fluorescence signal after binding with N-methyl-mesoporphyrin IX (NMM). The proposed sensing strategy can detect OTA with concentration as low as 4.9 pM. Besides, satisfactory results have also been obtained in the tests of actual samples. More importantly, the thermodynamic properties of nucleic acid chains in the monitoring platform were analyzed and the reaction processes and conditions were simulated before carrying out biological experiments, which theoretically proved the feasibility and simplified subsequent experimental operations. Therefore, the proposed method possess a certain application value in terms of monitoring mycotoxins in food samples and improving the quality control of food security.

## 1. Introduction

Ochratoxin A (OTA), another major contaminating mycotoxins in various food commodities such as grains, vegetables, nuts, spices, wine and animal feed [1,2], has attracted worldwide attention after aflatoxin [3]. The mycotoxin has highly kidney and liver toxicity, teratogenicity, carcinogenicity, mutagenicity and immunosuppressive effects on animals and humans [4,5,6]. As a consequence, many international agencies have specified the highest levels of OTA in different foodstuffs. For OTA in wine and grape-based beverages, the European Commission has set the maximum levels at 2 µg/kg. For unprocessed cereals, cereal-derived products and dried vine fruit, the levels were set at 5 µg/kg, 3 µg/kg and 10 µg/kg, respectively [7]. However, modern food processing technology cannot solve OTA contamination which can even survive in commercialized food systems such as bread, dried fruits, wine, and meat products, on account of its long half-life. Once ingested by human bodies, it will persist internally for more than 35 days [8,9]. Hence, it is essential to quantitatively detect OTA in foodstuffs for food control and human health. 

The official methods for detecting OTA include thin-layer chromatography (TLC) [10], high performance liquid chromatography (HPLC) [11,12], high performance liquid chromatography–tandem mass spectrometry (HPLC–MS/MS) [13], gas chromatography (GC) [14], total internal reflection ellipsometry (TIRE) [15,16]. Although these methods show high sensitivity and low detection limits, they still need expensive and time-consuming pretreatments such as extraction, sample clean up and preconcentration. Such analytical processes are laborious and require advanced equipment and well-trained laboratory personnel. Immunoassays based on antigen-antibody interactions are also widely used in OTA detection in foodstuffs, such as enzyme linked immunosorbent assays (ELISA), surface plasmon resonance (SPR) and electrochemical immunosensors [17,18,19,20]. These assays show high sensitivity of detection, but often suffer from cross-reactively, matrix interference and poor shelf life of antibodies. Moreover, the production of antibodies are expensive and time-consuming which may take several weeks. Therefore, there is still a need to establish a simple, economical and reliable detection platform.

Aptamers, a sequence of oligonucleotides obtained by repeated screening from the library of random oligonucleotide sequences are synthesized artificially in vitro using the ligand index enrichment system evolution (SELEX) technique [21]. They can be used to identify different target elements, such as drugs, proteins, small molecules and cells [22] due to the characteristics of high affinity, miraculous selectivity, good thermal stability and easy to synthesize. Since the OTA aptamer has been reported [23], a great deal of aptamer-based biosensors for detecting OTA have been widely developed. Besides, many signal amplification techniques have also been reported to improve the sensitivity and dynamic detection range, including colorimetric and optical determination based on nanoparticles [24,25], nicking enzyme-assisted fluorescence signal enhancement [26,27], electrochemical impedance spectroscopy based on gold nanoparticles [28,29], fluorescence determination based on dye labeling [30], etc. These methods can greatly reduce the detection limit of the biosensors, but unfortunately, the use of nanomaterials can be disturbed by other electroactive substances coexisting in actual samples [31]. The protein enzyme’s activity depends largely on the actual reaction conditions [32]. The preparation of gold nanoparticles and the fabrication of electrochemical biosensors are complicated and time-consuming [33]. In addition, the aptamers modified with chemical groups are expensive and may reduce the binding force between the aptamers and the molecular targets [34]. Therefore, it is necessary to develop a non-enzyme and label-free signal amplification method in order to detect OTA sensitively.

Hybridization chain reaction (HCR) is an isothermal, non-enzyme signal amplification technique originally proposed by Dirks and Pierce in 2004 [35]. In HCR, once the target DNA is introduced, two synthetic DNA hairpins coexisting in solution will hybridize into a continuous DNA nanowires. Due to the significant advantages of high amplification, controlled kinetics and non-enzyme natures, it has been widespread applied in the detection of proteins, nucleic acids and small molecules [36,37,38]. Nevertheless, most HCR-based detection methods require the modification of fluorophores and quenching groups, which increases the cost of detection and fluorescent background. Considering the above reasons, we introduced a unique high-order structure, G-quadruplex, instead of chemical labelling. Under the action of special cationic dyes, G-rich DNA sequences will pile up together to form G-quadruplex structure [39]. It can combine with NMM, which is a commercially available asymmetric anionic porphyrin that can specifically recognize the G-quadruplex structure rather than single-, duplex-, or triplex-stranded nucleic acid structure. After binding to the G-quadruplex, it shows a >20-fold increase in its fluorescence [40,41].

Actually, computer technology can solve the time-consuming, expensive and cumbersome shortcomings of biological experiments through simulation. Among them, by estimating the minimum free energy (MFE) of nucleic acids structures, the biological function of the relevant nucleotide sequence and the complete tertiary structure in the organism can be simulated and predicted [42]. Besides, the nucleotide sequences and reaction processes can be constructed by computer to simulate the corresponding nucleic acid models and perform related tasks dynamically. Therefore, we creatively combine computer simulation and biological experiments in our work for the sake of experiment time benefit and model execution efficiency. 

To achieve non-enzyme and non-label sensitive detection of OTA in agricultural products, we use G-quadruplex/NMM as the fluorescent signal reporter gene, and use HCR to further amplify the fluorescent signal. More importantly, we performed a series of computer simulation analysis before the biological testing to simplify the subsequent experimental steps and eliminate some negative effects. In the presence of OTA, the two designed DNA hairpins will sequentially open up and self-assemble into continuous DNA duplexes embedded with numerous G-quadruplexes [43], thereby significantly enhancing the fluorescent signal. Our method displays high sensitivity, less time consumption, strong selectivity and has certain practical application towards OTA detection, which can open up new approaches for the use of aptasensor in the fields of food control and quality inspection.

## 2. Result and Discussion

### 2.1. Mechanism for OTA Detection

The designed biosensing platform for non-enzyme and non-label OTA fluorescence detection is shown in Figure 1. The biosensor model contains two hairpin probes (H1, H2), both of which have six nucleotide (nt) sticky ends. In particular, the 5′ end of hairpin probe H1 has OTA aptamer sequences (rose red and green, Figure 1) and the loop portion of hairpin probe H2 contains G-quadruplex sequence (blue, Figure 1). The same colored sections in H1 and H2 indicate that the sequence are complementary. After the introduction of OTA, due to the specificity and high affinity of aptamer and target, it will combine with the hairpin probe H1 to form OTA-aptamer complex, thereby opening the stem-loop structure of H1 and exposing the foothold region. Subsequently, the exposed part of H1 (3′ end) will hybridize to the longer end of H2 (3′ end). After opening the hairpin probe H2, the exposed 5′ end of H2 can open up hairpin H1 again. Therefore, the hairpin probes are sequentially opened and assembled into continuous DNA duplexes. Among them, G-rich sequences will shape numerous G-quadruplexes under the action of K^+^. Finally, the fluorescent signals can be significantly enhanced by interact with NMM. In contrast, in the absence of OTA, the mixed solution of H1 and H2 merely showing a relatively low background fluorescence signal.

### 2.2. Verification of Feasibility by Computer Simulation and Biological Experiments

Before conducting actual experiments, it is essential to carry out computer simulations of the method proposed in order to economize time and cost of subsequent operations. Since nucleic acid secondary structure is critical to the function of the nucleic acid strands, we introduced an algorithm to predict and analyze the thermodynamic properties of the strands designed in this paper. In general, the combination of thermodynamic models and dynamic programming algorithms can estimate the minimum free energy (MFE) of nucleic acid structures with different loops and calculate the partition function [44]. MFE can be applied to predict the thermal stability of DNA and RNA strands [45] and partition function plays a major role in evaluating DNA and RNA sequences designed in the conformational ensemble.

Specifically, in the absence of a pseudoknot, thermodynamic model decompose the secondary structures of DNA and RNA molecules into different loops based on the base-pairing diagram. These loop configurations are associated with entropy and enthalpy values measured from loop sequence, type, and length [46]. Beginning with the study of Tinoco [47], numerous researchers have worked on the physical models of these structures. As shown in the base-pairing diagram of Figure 2, the recognized loop types include an interior loop, hairpin loops, a bulge loop, a multiloop and stacked bases. Meanwhile, the polymer main chain is represented by a straight line in the polymer graph, and the complementary paired bases are linked by arcs. All loop structures are nested with no crossing arcs. Furthermore, the free energy of a secondary structure S is vitally interrelated to the free energy F_L_ of each loop L it contains, so the total free energy F(S) can be calculated in Equation (1). The additivity of free energy means stronger impact on the partition function Q defined by Equation (2). Afterwards, the equilibrium probability of any nucleic acid secondary structure S can be calculated by weights (Equation (3)), where T and R represent temperature and universal gas constant.
(1)$$F(S)=\sum_{L\in S}F_{L}$$
(2)$$Q=\sum_{S}e^{-[F(S)/\mathrm{TR}]}$$
(3)$$P(S)=\frac{1}{Q}e^{-[F(S)/\mathrm{TR}]}$$

According to the above calculation methods, the MFE and secondary structure of the hairpin structures H1 and H2 designed in this model were estimated and simulated separately. It can be seen from Figure 3 that the two single strands were spontaneously folded into expected hairpin structures at 37 °C with relatively low free energy (F(H1) = −16.20 kcal/mol, F(H2) = −10.75 kcal/mol) by NUPACK simulation [48]. The lower free energy, the more stable structure. The results theoretically illustrate the feasibility of our design of the two hairpin probe sequences.

Moreover, in order to test whether the reaction meets the expectation and further simplify biological experiments, the experimental process, chain concentration and products were simulated and optimized by Visual DSD [49]. First of all, we processed the OTA into a single chain to facilitate computer input, which was complementary to its aptamer. It is worth noting that the HCR reaction products are long DNA nanowires, so we set reactants H1 and H2 forming the DNA duplex structures to a lower concentration than actual experiments for the convenience of computer output. In the presence of OTA (Figure 4a), due to the strong interaction between the aptamer in hairpin probe H1 and OTA, the concentration of OTA (Figure 4a, yellow curve) decreased rapidly to form OTA-H1 complexes, and eventually tends to 0. At the same time, the opened hairpin probe H1 can bind to H2 to shape Duplex 1 (Figure 4a, rose red curve). Afterwards, the exposed footholds of H2 could open the hairpin structure of H1 again, so that Duplex 1 (OTA-H1-H2) quickly disappeared and evolved into Duplex 2 (OTA-H1-H2-H1), which will immediately combined with new H1 into Duplex 3 (OTA-H1-H2-H1-H2). Therefore, the concentration of H1 and H2 gradually decreased during the reaction (Figure 4a, blue curve, red curve, respectively) and the duplexes were continuously produced and rapidly disappeared for evolving into longer DNA nanowires on account of constant hybridization between H1 and H2. In contrast, in the absence of OTA (Figure 4b), the concentration of hairpin probes H1 and H2 remained unchanged (Figure 4b, blue curve, red curve, respectively), indicating that no various duplexes were engendered. As we can see, the above results confirmed that our proposed strategy was theoretically feasible. At the same time, the experimental process can be simulated by changing the concentration and ratio of diverse reactive substances and the time and conditions of reactions, thereby greatly simplify subsequent actual experimental operations.

Subsequently, biological experiments were conducted to validate the practical feasibility of the proposed method for OTA detection. The fluorescence intensity changes of different solutions were recorded in Figure 5. The solution containing only OTA and H1 (curve c) or H2 (curve d) exhibited a relatively low fluorescence value. In the absence of OTA, H1 and H2 remained stable and the solution only showed a negligible change in fluorescence intensity (curve b). Then, adding OTA to test tubes containing H1 and H2 strands increased the fluorescence intensity significantly (curve a). These results are consistent with the demonstration in Figure 4, proving the feasibility of our model for OTA detection.

### 2.3. Optimization of OTA Detection Conditions

For the sake of obtaining optimal performance of the proposed detection platform, several important parameters were optimized on the basis of single-factor experiments. Firstly, the effect of K^+^ concentration on the changes of fluorescence intensity was studied because the sequences of G-rich oligonucleotide can form stable G-quadruplex structures under the action of K^+^. Furthermore, the K^+^ can mutual coordinate with carbonyl oxygen atoms of the G-residues and be embedded in the central of two stacked G-tetrads [36]. As shown in Figure 6a, F_0_ and F represent the measured fluorescence value at 608 nm before and after adding OTA, respectively. With the increase of K^+^ concentration, the amount of fluorescence intensity change (F-F_0_) gradually increased and reached to the maximum at 20 mM, indicating that 20 mM of K^+^ could completely accelerate the folding of G-quadruplex structures. Therefore, 20 mM K^+^ concentration was chosen for the next experiment.

Subsequently, the HCR reaction time for H1 and H2 was also optimized. As the reaction time gradually increased in Figure 6b, the degree of hybridization between H1 and H2 deepened and the fluorescence change remained steady after 60 min, manifesting an equilibrium for HCR assembly between H1 and H2. Consequently, the time for HCR was set at 60 min in the subsequent experiments.

Moreover, the concentration of NMM and its reaction time with the G-quadruplexes produced in the experiment also directly affect the fluorescence intensity of the solution. According to Figure 6c, 1.5 μM of NMM displayed the highest fluorescence change in the detection because the lower concentration of NMM cannot provide sufficient fluorescence intensity for the G-quadruplexes produced in the reaction, and the higher concentration of NMM may engender increased background fluorescence signal. It was worth noting that the incubation time with NMM had little impact on the variation of fluorescence intensity (Figure 6d). Hence, in order to save the general time for the experiment, 10 min was selected as the combination time with NMM.

### 2.4. Sensitivity and Specificity

According to the optimal experimental conditions obtained from the above single-factor experiments, the sensitivity of the biosensor can be further analyzed by detecting the fluorescence intensity of different concentrations of OTA. As shown in Figure 7a, increasing OTA from 0.01 nM to 50 nM results in a gradual enhance in fluorescence signal. In addition, the fluorescence value of NMM at 608 nm is proportional to the logarithm of the OTA concentration from 0.01 nM to 0.5 nM (Figure 7b). The linear regression equation is y = 457.535lgx + 620.267 (x and y refer to OTA concentration and fluorescence intensity, respectively) with the correlation coefficient of 0.9948. Furthermore, the calculated detection limit (LOD) for OTA is 4.9 pM (according to 3σ/S rule). When the OTA concentration of the detected solution is above the linear range (i.e., 0.5 nM), by simply diluting the actual samples to a calculable concentration with buffer solution, the target concentration of OTA can be estimated quantitatively according to the multiple of dilution. 

Compared with other proposed strategies for OTA detection (Table 1), although our method is not as sensitive as electrochemical and immunofluorescence assays, the platform is economical, convenient and fast, only takes one and a half hours from preparation to detection. In addition, this work uses non-enzyme and non-label strategies and has a lower detection limit compared with general colorimetry, fluorescence and chemiluminescence methods.

In order to verify the specificity of the developed sensor, other mycotoxins (OTB and AFB1) were also tested under the same conditions. According to Figure 8, even if there were other mycotoxins with concentrations ten times higher than OTA, negligible changes in fluorescence intensity could be observed. Nevertheless, the presence of OTA caused significant increase in fluorescence intensity. In addition, the fluorescence signal of the mixture of OTA and other control mycotoxins was similar to that of the OTA group. The results suggest that this method possesses an outstanding specificity of OTA detection.

### 2.5. Application in Practical Samples

Subsequently, the practical application potential of the fluorescent aptasensor was evaluated by adding three different concentrations of OTA in actual wheat flour samples through standard addition methods to determine recovery rates. Table 2 shows that the recoveries determined in wheat flour samples were between 97.9% and 105%, and the relative standard deviation (RSD) was lower than ±4.8%. In addition, the detected recovery of three wine samples was higher than 94% and the RSD in the range from 3.7% to 5.1%. The experimental results suggest that our approach may be an effective and convenient method for OTA detection in actual agricultural commodities, and may provide promising strategies for improving food quality and safety.

## 3. Conclusions

In short, combining with the target-triggered structure-switching signaling aptamer and HCR technology, we have proposed a non-enzyme and non-label fluorescence biosensing system for OTA detection, proved by computer simulations and biological experiments. The approach we developed exhibits several advantages. First, the biosensor model has specific recognition and awesome detection ability for OTA with no modification of fluorescent groups and quenching groups, making the experiments more economical and unsophisticated. Second, by conducting computer simulations to verify and optimize the experimental process, making the subsequent biological experiments more facile and effective. Third, using HCR for signal amplification rather than other proteases and complex thermal cycling processes makes the operation more convenient and controllable with an OTA detection down to 4.9 pM. Furthermore, the detection and analysis of other interfering mycotoxins and actual samples showed that the proposed sensor system possesses high specificity and practical application potential. Finally, the strategy has good universal adaptability for detecting other small molecules and proteins by skillfully designing aptamer sequence of the hairpin probe. In summary, due to its simple design and operation, high sensitivity and specificity, and low cost, we anticipate that the platform could open up new opportunities for the detection of mycotoxins and contaminants in foodstuffs, and provide new ideas in future research of the interdisciplinary discipline of biology and computers.

## 4. Materials and Methods

### 4.1. Reagents

OTA, ochratoxin B (OTB), ochratoxin C (OTC) and aflatoxin B1 (AFB1) were purchased from Pribolab Co., Ltd. (Qingdao, China). N-methyl-mesoporphyrin IX (NMM) was bought from JKchemical Co., Ltd. (Beijing, China), stored at −20 °C in the dark before use. The ssDNA used in the experiments were synthesized and further purified by Sangon Biotech Co., Ltd. (Shanghai, China). The sequence of ssDNA (H1) is 5′-GAT CGG GTG TGG GTG GCG TAA AGG GAG CAT CGG ACA CGC CAC CCA CAC-3′, where the italic sequences are OTA aptamer. The sequence of ssDNA (H2) is 5′-CCA CAC CCG ATC CTG GGA GGG AGG GAG GGG TGT GGG TGG CG-3′, and some of which can form G-quadruplex structure. By dissolving the DNA strands in 10 mM Tris buffer (200 mM NaCl, 20 mM MgCl_2_, pH 7.4) to obtain the DNA stock solution, and then stored at −4 °C before use. The ultra pure water used in our experiment was purified by a Milli-Q system (18 MΩ cm). 96-Well microplates were purchased from Lingyi Biotech Co., Ltd. (Shanghai, China).

### 4.2. Instrumentation

The fluorescence spectrum of NMM was gauged by fluorescence scanning spectrometer under 399 nm excitation and 610 nm emission through EnSpire ELIASA from PerkinElmer USA company (Shanghai, China).

### 4.3. Fluorescent Detection of OTA 

Before the experiments, the diluted solution with H1 (1 μM) and H2 (1 μM) were heated at 95 °C by PCR for 5 min, then cooled to room temperature slowly at a rate of 1 °C/min to form a hairpin structure. Subsequently, different concentrations of OTA, H1 (300 nM), KCl (20 mM) were mixed with working buffer and heated at 37 °C for 15 min. H2 (300 nM) was then put in with a pipette and the reaction was continued at 37 °C for 60 min. After self-assembly was completed, NMM (1.5 μM) was added and further incubated for 10 min at 25 °C. Finally, a part of the solution was removed to a 96-well plate, and the fluorescence intensity was detected and recorded by the instrument mentioned above.

### 4.4. OTA Detection in Wheat Flour and Red Wine Samples

Wheat flour and red wine samples were purchased from local supermarkets. Mix aliquots of wheat flour (1 g), different concentrations of OTA and 10 mL of extraction solvent (methanol: water, 6:4 (v/v)) in a vortex mixer for 5 min. After centrifuging for 10 min, the supernatant was removed and diluted with buffer, and then used for detection. At the same time, by adding different concentrations of OTA to the solution containing 1% red wine to prepare three samples with OTA concentration of 10 pM, 50 pM, and 200 pM respectively. All samples were then detected and analyzed as described above.

## Figures and Tables

**Figure 1 toxins-12-00376-f001:**
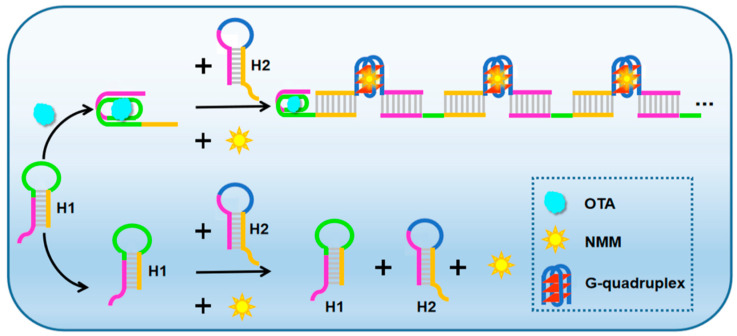
Schematic diagram for OTA detection based on HCR and G-quadruplex structures.

**Figure 2 toxins-12-00376-f002:**
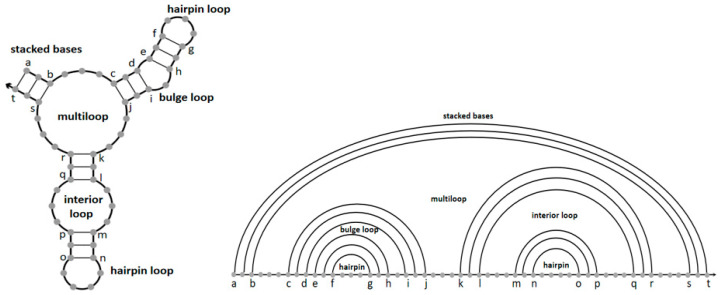
Canonical loop types of nucleic acid structure.

**Figure 3 toxins-12-00376-f003:**
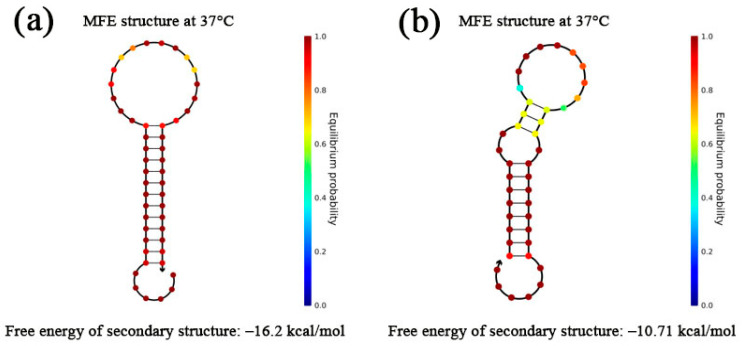
Simulation results of MFE structure at 37 °C of H1 (**a**) and H2 (**b**).

**Figure 4 toxins-12-00376-f004:**
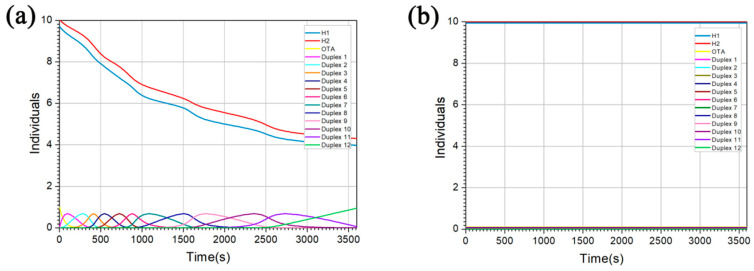
Visual DSD simulations. Changes in the concentration of various chains over time under different conditions: (**a**) with OTA; (**b**) without OTA.

**Figure 5 toxins-12-00376-f005:**
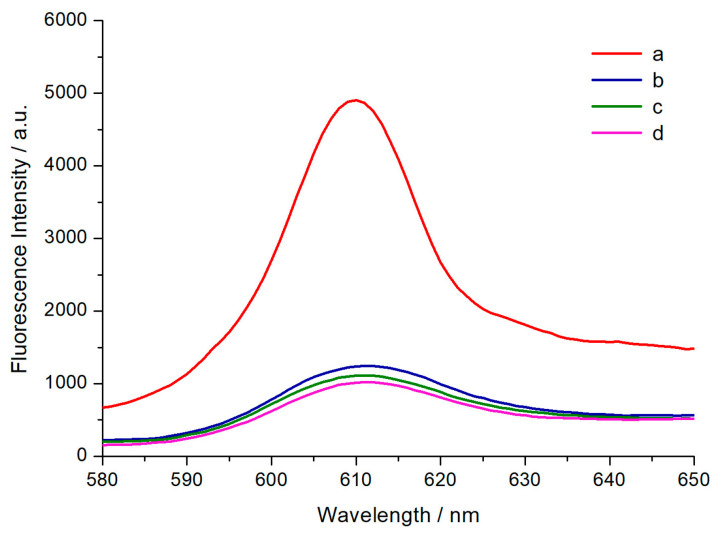
Fluorescence spectra of NMM with different substances: (**a**) H1, H2 and OTA; (**b**) H1 and H2; (**c**) H1 and OTA; (**d**) H2 and OTA. Experimental conditions: [H1] = [H2] = 300 nM, [OTA] = 10 nM.

**Figure 6 toxins-12-00376-f006:**
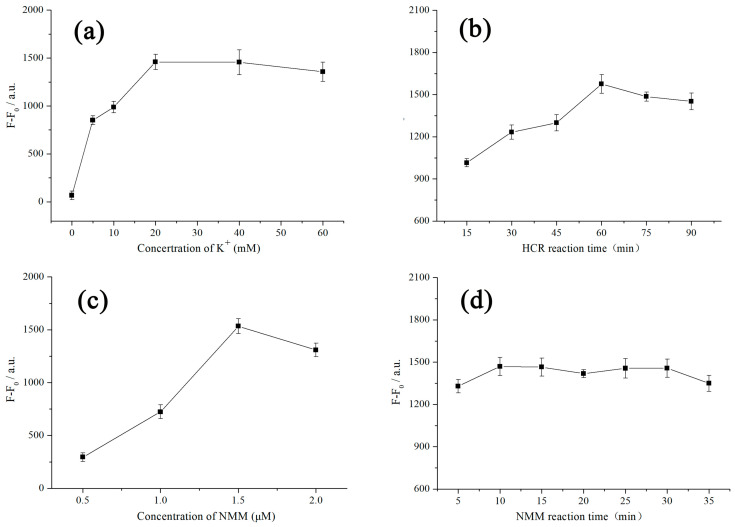
The fluorescence intensity change at 608 nm versus: (**a**) the concentration of K^+^. [NMM] = 1.5 μM, and the reaction time of HCR and NMM are 60 min and 10 min, respectively; (**b**) the reaction time of HCR. [NMM] = 1.5 μM, [K^+^] = 20 mM, and the combination time of NMM is 10 min; (**c**) the concentration of NMM. [K^+^] = 20 mM, and the reaction time of HCR and NMM are 60 min and 10 min, respectively; (**d**) the reaction time of NMM. [NMM] = 1.5 μM, [K^+^] = 20 mM, and the reaction time of HCR is 60 min. Other conditions: [H1] = [H2] = 300 nM. Error bars, SD, *n* = 3.

**Figure 7 toxins-12-00376-f007:**
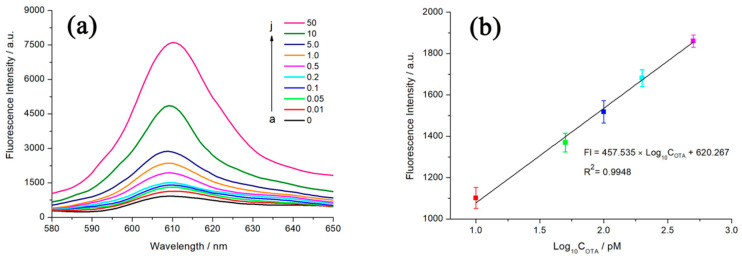
(**a**) Fluorescence spectra of different concentrations of OTA. From a to j, the concentrations of OTA is 0, 0.01, 0.05, 0.1, 0.2, 0.5, 1.0, 5.0, 10, 50 nM, respectively; (**b**) Linear relationship between the fluorescence value of (**a**) at 608 nm versus logarithmic concentration of OTA. Error bars, SD, *n* = 3.

**Figure 8 toxins-12-00376-f008:**
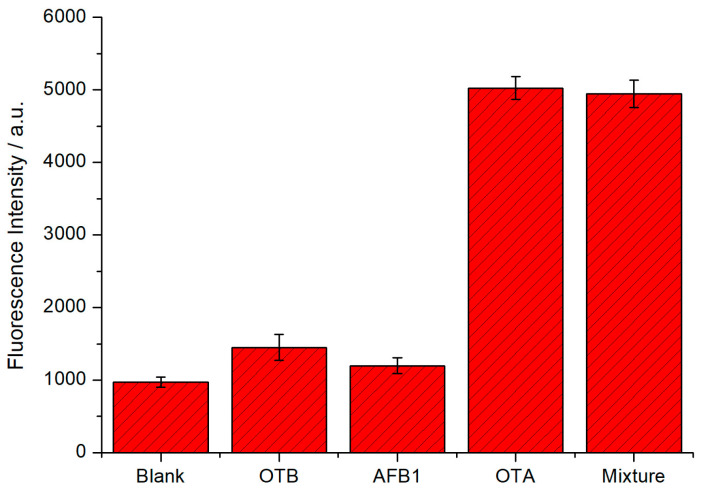
Specificity of the fluorescent biosensor to OTA other mycotoxins and the mixture of OTA, OTB and AFB1. The concentration of OTA is 10 nM and other mycotoxins are 100 nM. Error bars, SD, *n* = 3.

**Table 1 toxins-12-00376-t001:** Comparison of proposed OTA detection strategies and this work.

Detection Method	Matrix	LOD	References
colorimetric	peanuts, corn	74.3 pM	[21]
fluorescence	bear, red wine	4.2 nM	[23]
fluorescence	red wine	198.1 pM	[24]
electrochemical	soybean	5.2 fg mL^−1^	[26]
immunofluorescence	corn, rice, wheat	0.12 pM	[50]
fluorescence	corn flour	30 pM	[51]
chemiluminescence	wheat, rice, core	10.6 pM	[52]
chemiluminescence	coffee	0.5 nM	[53]
electrochemical	coffee	0.125 ng mL^−1^	[54]
fluorescence	wheat flour, red wine	4.9 pM	this work

**Table 2 toxins-12-00376-t002:** Application of fluorescent aptamer sensor for OTA determination in wheat flour and red wine samples.

Samples		Added (pM)	Found (pM)	Recovery (%) ^a^	RSD (%) ^b^
Wheat flour	1	10	10.5	105	±4.1
	2	50	48.9	97.9	±3.3
	3	100	104.8	104.8	±4.8
Red wine	1	10	9.4	94	±3.7
	2	50	48.4	96.8	±5.1
	3	100	95.7	95.7	±4.9

^a^ The mean of three measurements. ^b^ RSD = The relative standard deviation.

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
