# Peer review of "A Non-Enzyme and Non-Label Sensitive Fluorescent Aptasensor Based on Simulation-Assisted and Target-Triggered Hairpin Probe Self-Assembly for Ochratoxin a Detection"

_toxins, 2020, doi:10.3390/toxins12060376_

Round 1

Reviewer 1 Report

Title: A Non-Enzyme and Non-Label Sensitive Fluorescent Aptasensor Based on Simulation-Assisted and 3 Target-Triggered Hairpin Probe Self-Assembly for Ochratoxin A Detection

Article Type: Article

I believe that this manuscript is a good contribution to research in this area. 

In the text of the manuscript, the authors can emphasize the importance of using this method a non-enzyme and non-label sensitive fluorescent aptasensor in comparison with other general analytical methods.

The objectives of this study were carried and are significant for further research.

Comments:

Page 2, line 40-42 …In general methods for detecting OTA was not provided LC/MS/MS including a recent reference!

The technological methodology, the design of the study and determination and quantification by a non-enzyme and non-label sensitive fluorescent aptasensor are appropriate, but the following references were not provided and discussed in the manuscript:

1.

Goud, K.Y., Reddy, K.K., Satyanarayana, M. et al. (2020). A review on recent developments in optical and electrochemical aptamer-based assays for mycotoxins using advanced nanomaterials. Microchim Acta 187, 29. https://doi.org/10.1007/s00604-019-4034-0

2.

Gu C, Yang L, Wang M, Zhou N, He L, Zhang Z, du M (2019) A bimetallic (cu-co) Prussian blue analogue loaded with gold nanoparticles for impedimetric aptasensing of ochratoxin a. Microchim Acta 186:343. https://doi.org/10.1007/s00604-019-3479-5

3.

Zejli, K. Yugender Goud, Jean Louis Marty (2018). Label free aptasensor for ochratoxin A detection using polythiophene-3-carboxylic acid, Talanta, Volume 185, Pages 513-519. https://doi.org/10.1016/j.talanta.2018.03.089.

4.

Wei M, Zhang W (2017). A novel impedimetric aptasensor based on AuNPs–carboxylic porous carbon for the ultrasensitive detection of ochratoxin A. RSC Adv 7:28655–28660. https://doi.org/10.1039/C7RA04209D

Author Response

Dear Reviewer,

Thank you for giving us constructive suggestions which would help us both in English and in depth to improve the quality of the paper. The following is a point-to-point response to your comments.

Point 1: Page 2, line 40-42 …In general methods for detecting OTA was not provided LC/MS/MS including a recent reference!

Response 1: Thank you for the comments on the paper. We’ve added other offcial OTA detection methods in line 40-42 and line 47 such as thin-layer chromatography (TLC), high performance liquid chromatography-Tandem mass spectrometry (HPLC–MS/MS) and surface plasmon resonance (SPR). Besides, in line 335, 339, 344 and 347 we have also added  references to the corresponding detection methods  for the past two years.

Point 2: The technological methodology, the design of the study and determination and quantification by a non-enzyme and non-label sensitive fluorescent aptasensor are appropriate, but the following references were not provided and discussed in the manuscript...

Response 2: Your suggestion is greatly appreciated. We found that there is no mention of electrochemical method in our manuscript. According to your suggestion, we added your suggested method in line 60-61 (colorimetric and optical determination based on nanoparticles [21,22], nicking enzyme-assisted fluorescence signal enhancement [23,24]). In addition, the comparison with other proposed strategies in Table 1 also added the comparison of electrochemical methods. At the same time, we also discussed the merits and demerits of using nano materials and electrochemical methods. (Line 62-67: Although methods can greatly reduce the detection limit of the biosensors, the use of nanomaterials can be disturbed by other electroactive substances coexisting in actual samples. The preparation of gold nanoparticles and the fabrication of electrochemical biosensors are complicated and time-consuming.)

Point 3: In the text of the manuscript, the authors can emphasize the importance of using this method a non-enzyme and non-label sensitive fluorescent aptasensor in comparison with other general analytical methods.

Response 3: Thank you for your valuable advice.

For OTA detection methods: we analyzed the advantages and disadvantages of official methods and immunoassays. (Line 42-50: Such analytical processes are laborious and require advanced equipments and well-trained laboratory personnel, ... , and often suffer from cross-reactively, matrix interference and poor shelf life of antibody...) 

For signal amplification techniques: we also analyzed several signal amplification methods (Reference 21-27), and discussed merits and demerits of using nano materials, protein enzymes, fabrication of electrochemical, aptamers modified with chemical groups  (Line 62-68 ).  

For the detection results: we compare the detection limits of several detection methods and our work in Table 1 (Line 238) and discuss them in line 233-237. The electrochemical and immunofluorescence assays have lower detection limits, but our method is economical, convenientand and fast, only takes one and a half hours from preparation to detection. Meanwhile, it has a lower detection limit compared with general colorimetry, fluorescence and chemiluminescence methods.

Reviewer 2 Report

This article describes the development of a new quantitative detection method for Ochratoxin A (OTA), a concerning, high half-life mycotoxin for which maximum levels in food are already in place. The authors aimed to present a simple, economical and reliable detection, without drawbacks like expensive equipment, time-consuming pretreatments, poor antibody shelf life and cross-reactivity faced by established HPLC, GC or ELISA methods. Therefore, a G-quadruplex high order structure (formed from G-rich DNA), able to combine with NMM to yield 20-fold fluorescence enhancement, was used as the fluorescent signal reporter gene. By using the non-enzyme and label-free hybridization chain reaction as a further signal amplification for the already existing OTA aptamer, the authors aimed to avoid problems arising from alternative methods. To identify the right nucleotide sequences and to cut back on biological experiments, computer simulations were utilized.

In the results and discussion section, the OTA detection based on HCR and G-quadruplex structures and the use of computer simulations to predict thermodynamic properties of the nucleic strands and to model the reactions, is explained in a detailed and comprehensive manner.

The optimization of OTA detection conditions was done by biological experiments and includes the K+ concentration, reaction time, NMM concentration and reaction time of NMM. Furthermore, sensitivity and specificity were investigated. It was discovered that quantitative analysis could be achieved by dilution experiments and that the specificity against other possible mycotoxin contaminants is sufficient. Recovery rates from actual samples by standard addition method were also good.

All in all, the presented work proposes a new, innovative method for the detection of OTA. The fluorescence signal amplification by HCR is a smart way to possibly circumvent problems that alternative methods face. Though, since there are already several alternative methods published, a definitive answer to which method is the best, most economical, less time consuming, reliable etc. is not possible from this work. So, statements from the authors to this might be overstated. For this method to come out on top as the go to, convenient and practical method for OTA (and possible other analytes) analysis, a direct comparison study of various methods would be necessary in the future.

Nevertheless, the content of the paper is promising, quite innovative and should definitely be published. Especially the utilization of computer simulations is impressive. The presented signal enhancement approach may very well be interesting to use for other analytes and therefore importance of this work is high.

Orthography:

Line 46-47: Aptamers, a sequence of oligonucleotides obtained by repeated screening from the library of random oligonucleotide sequences [are] synthesized artificially in vitro using the ligand index enrichment system evolution (SELEX) technique.

Line 123: show[n]

Author Response

Dear Reviewer,

Thank you for giving us constructive suggestions which would help us both in English and in depth to improve the quality of the paper. The following is a point-to-point response to your comments.

Point 1:  Though, since there are already several alternative methods published, a definitive answer to which method is the best, most economical, less time consuming, reliable etc. is not possible from this work. So, statements from the authors to this might be overstated. For this method to come out on top as the go to, convenient and practical method for OTA (and possible other analytes) analysis, a direct comparison study of various methods would be necessary in the future.

Response 1: Your suggestion is greatly appreciated. According to your suggestion, we have added three parts to the paper.

For OTA detection methods: we analyzed the advantages and disadvantages of official methods (TLC, HPLC, HPLC–MS/MS, GC) and immunoassays (ELISA, SPR, electrochemical immunosensors) of OTA detection, and discussed them in line 42-50 (Although these methods show high sensitivity and low detection limits...).

For signal amplification techniques: we also analyzed several signal amplification methods (Reference 21-27), and discussed the merits and demerits of using nano materials, protein enzymes, fabrication of electrochemical, aptamers modified with chemical groups  (Line 62-68).  

For the detection results: we compare the detection limits of several detection methods with our work in Table 1 (Line 238) and discuss them in line 233-237. The electrochemical and immunofluorescence assays have lower detection limits, but our method is economical, convenientand and fast, only takes one and a half hours from preparation to detection.

Point 2: Line 46-47: Aptamers, a sequence of oligonucleotides obtained by repeated screening from the library of random oligonucleotide sequences [are] synthesized artificially in vitro using the ligand index enrichment system evolution (SELEX) technique. 

Line 123: show[n]

Response 2: Thank you for your careful work. We are sorry for these mistakes and have corrected these grammatical errors on Line 53 and Line 134.

Round 2

Reviewer 1 Report

Manuscript Number: toxins-814777 revision

Title: A Non-Enzyme and Non-Label Sensitive Fluorescent Aptasensor Based on Simulation-Assisted and 3 Target-Triggered Hairpin Probe Self-Assembly for Ochratoxin A Detection

Article Type: Article

Comments:

I don´t have cardinal objections. 

Author Response

Dear Reviewer,

Thank you again for your comments and suggestions on our manuscript and wish you all the best.